# Entropy and Fractal Techniques for Monitoring Fish Behaviour and Welfare in Aquacultural Precision Fish Farming—A Review

**DOI:** 10.3390/e25040559

**Published:** 2023-03-24

**Authors:** Harkaitz Eguiraun, Iciar Martinez

**Affiliations:** 1Department of Graphic Design & Engineering Projects, Faculty of Engineering in Bilbao, University of the Basque Country UPV/EHU, 48013 Bilbao, Bizkaia, Spain; 2Research Center for Experimental Marine Biology and Biotechnology—Plentziako Itsas Estazioa (PiE-UPV/EHU), University of the Basque Country (UPV/EHU), 48620 Plentzia, Bizkaia, Spain; 3Department of Zoology and Animal Cell Biology, Faculty of Science and Technology, University of the Basque Country (UPV/EHU), 48940 Leioa, Bizkaia, Spain; 4IKERBASQUE, Basque Foundation for Science, 48009 Bilbao, Bizkaia, Spain

**Keywords:** entropy, fractal dimension, fish behaviour, fish welfare, precision fish farming, intelligent aquaculture, pain, fear/anxiety, positive emotional contagion, hierarchies

## Abstract

In a non-linear system, such as a biological system, the change of the output (e.g., behaviour) is not proportional to the change of the input (e.g., exposure to stressors). In addition, biological systems also change over time, i.e., they are dynamic. Non-linear dynamical analyses of biological systems have revealed hidden structures and patterns of behaviour that are not discernible by classical methods. Entropy analyses can quantify their degree of predictability and the directionality of individual interactions, while fractal dimension (FD) analyses can expose patterns of behaviour within apparently random ones. The incorporation of these techniques into the architecture of precision fish farming (PFF) and intelligent aquaculture (IA) is becoming increasingly necessary to understand and predict the evolution of the status of farmed fish. This review summarizes recent works on the application of entropy and FD techniques to selected individual and collective fish behaviours influenced by the number of fish, tagging, pain, preying/feed search, fear/anxiety (and its modulation) and positive emotional contagion (the social contagion of positive emotions). Furthermore, it presents an investigation of collective and individual interactions in shoals, an exposure of the dynamics of inter-individual relationships and hierarchies, and the identification of individuals in groups. While most of the works have been carried out using model species, we believe that they have clear applications in PFF. The review ends by describing some of the major challenges in the field, two of which are, unsurprisingly, the acquisition of high-quality, reliable raw data and the construction of large, reliable databases of non-linear behavioural data for different species and farming conditions.

## 1. Introduction

Aquaculture is expected to play a key role in supplying high-value protein and micronutrients [1] to a human population estimated to reach about 9700 million by 2050 (United Nations World Population Prospects https://population.un.org/wpp/, accessed on 21 March 2023). Indeed, in Asia and some African countries, seafood already makes up over 50% of the animal protein in their diets [1]. Although most seafood is expected to originate from aquaculture [1], the fish farming industry faces some serious challenges, including climate change, environmental contaminants, and the need to ensure fish health and welfare, which will require the development of alternative disease treatments and the identification of novel sources of feed and nutrients [2]. In response to these challenges, a paradigm shift is mandatory [3]; thus, the European aquaculture industry [2] and European policy [4] are targeting green and resilient production systems and emerging technologies such as *Environmental Intelligence* and *Monitoring Systems*, both of which include *Precision Farming* as a key area [5]. Føre et al. [6] introduced the concept of Precision Fish Farming ((PF) which is comparable to Intelligent Aquaculture (IA)) to apply control-engineering principles to improve farmers’ ability to monitor, control, and document farm production. PFF consists of several cyclical operational processes performed in four phases [6]: Phases 1 and 2 constitute the observation and interpretation, respectively, of fishes’ responses, which form the basis for Phase 3, decision making, which, in turn, governs the actions implemented in Phase 4.

Observation requires the deployment of sensors and the implementation of monitoring methods, which were recently reviewed by [6,7]. The most common monitoring methods are based on the analysis of video images, while newer ones target sound/acoustic signals [8,9,10]. However, the sensors themselves may influence an animal’s behaviour; for instance, acoustic telemetry requires tagging [9,11], sonar affects a fish’s hearing [12], and the presence of robots or divers’ videorecording also alter fishes’ behaviour [13,14]. Fixed sensors interfere the least, but they can only record fish within a certain region. Phase 2, which is addressed in the present review, deals with the handling and interpretation of the large amount of data generated by automatic monitoring. However, the automatic interpretation of the data demands the application of machine learning, artificial intelligence (AI) algorithms, and decision support systems, whose addressal is outside the scope of the present work (see reviews by [7,15,16,17,18,19]). Phase 3 also benefits from AI algorithms and decision support systems, and both Phases 3 and 4 (implementation) will most likely ultimately rest on the farmer’s experience, and it will, in all probability, be the farmer, and not an automated system, who will make the ultimate decisions.

Understanding the status of fish and detecting the presence of stressors is important to improve animal welfare and optimize production yield and quality. Intensive fish farming includes all life stages from the broodstock to the marketable fish. The hatchery phase usually takes place indoors (and under completely controlled conditions), while the ongrowing phase usually occurs in recirculating aquaculture systems (RAS) or outdoor ponds or cages. Outdoor rearing exposes fish to predators, variable environmental conditions, contaminants (heavy metals, pesticides, drugs, etc.), diseases and infections (including parasites, bacterial and viral infections), micro- and nanoplastics, and anthropogenic noise pollution, which have all been shown to alter fish behaviour [20,21,22,23,24,25,26,27,28,29,30,31,32]. Changes in aquatic organisms’ behaviour have been proposed to serve as Biological Early Warning Systems (BEWS) to monitor both the presence of environmental contaminants in water resources [31,33,34] and the fish production in aquaculture [3,6,23,35]. We have recently reviewed the potential of entropy and fractal dimension (FD) analyses of individual and collective fish behaviours to establish Biological Early Warning Systems for the presence of environmental contaminants (Eguiraun and Martinez, submitted); therefore, it will not be addressed in the present work.

Unfortunately, there are numerous additional stressors that remain unknown or that are not easily measurable, such as social interactions. The existence of collective behaviour and potential social stressors must also be considered in both experimental settings and on farms. Fish have been shown to be truly social creatures with social structures and dominance hierarchies whose very complex and largely unknown formation [36] needs to be mapped on real-life farms to achieve optimal production. The fact that social structures and hierarchies under real farming conditions are largely unknown due to the difficulty of their documentation does not mean that they are irrelevant or that they do not influence the health, welfare, and other phenotypical/quality aspects of production. Methodologies for the fast detection of abnormal fish behaviours, which may be critical for the early detection of deficiencies during farming, have already been successfully tested applying a convolution 3D deep (C3D) network model optimized by cross-entropy loss to analyse real-life aquaculture video surveillance [37]. 

Laboratory studies with zebrafish have shown that the organizational patterns of dominance hierarchies are heavily influenced by individual interactions, indicating that fish systems are both self-structuring and self-organizing and that the evolution of behaviour in the formation of a dominance hierarchy is influenced by networks of individuals rather than independent interacting pairs of individuals ([38]; see [39] for a more general discussion on swarm behaviours). The formation of hierarchies is particularly interesting because it has been shown that too few fish in such a system (under about 15 individuals) will alter the normal behaviour of naturally gregarious species as shown by the system’s Shannon Entropy (SE) [35]. Studying the connectiveness of swarm behaviour, which can also be applied to fish shoals, Komareji et al. [39] identified a relationship between the size of the swarm/shoal (the number of individuals) and the number of topological nearest neighbours influencing any individual’s behaviour and dynamics. Importantly, the connectedness of the swarm and the structural properties of the swarm network are generally not constant, which is relevant if we wish to monitor the collective behaviour of fish on farms. A second important point is that the probability of the entire swarm being connected is a function of both the number of individuals in the swarm and the number of interacting nearest neighbours; thus, the larger the number of individuals, the larger the number of interacting nearest neighbours needed for the swarm to display a dynamic collective behaviour [39]. This means that to obtain information about production based on the collective behaviour of the fish system, the farming conditions must allow for the interaction (albeit not the necessarily direct interaction) of all the fish in the cage. In addition, the available space for the fish to swim must be proportional to the number of fish; for example, the interactions that occur when there are too few fish in a large cage may hinder the development of collective behaviour. 

Anthropogenic noise pollution is a less-studied source of stress for fish that negatively affects the physiology and behaviour of individuals as it induces temporary or permanent hearing loss, stress and behavioural reactions to noise [32], and changes in shoaling behaviour (discussed in the study by [25] and references therein). In addition, anthropogenic noise seems to increase the number of fish killed by predation [40]. The identification of stressful noises requires the use of passive acoustic monitoring devices to first identify the normal noises produced by the monitored species, followed by the noises precipitating the stress and, finally, the noises the fish produce in response to the stress. This approach is still in its infancy.

### 1.1. Applications of Entropy and Fractal Analyses to Fish Behaviour Studies

The interactions between individuals, including knowledge regarding who leads, i.e., “who follows whom”, is important in social species and requires the quantification of the direction of information flow between individuals. Transfer entropy (TE) is a method that is commonly used to identify the flow of information and thus detect leaders both in interactions between pairs of individuals and in groups. Zebrafish have often been used as a model system in studies using either real fish, robotic replicas, and/or mathematical models based on its behaviour [41]. However, it must be borne in mind that many of these studies are performed within a narrow time-window and assume that the leaders are consistent over time. Yet, this may not be the case since individuals in interacting pairs have been shown to swap roles occasionally and adjust their responses to one another as they exchange roles [42].

Non-linear dynamical analyses are becoming increasingly relevant for understanding biological systems. For instance, apparently random social and behavioural patterns in several species’ activities, including the swimming patterns of fish [43,44,45], have been shown to contain highly non-random components of a fractal nature (see the research conducted by [46] and references therein), i.e., they display self-similarity [47]. Likewise, the predictability of a system can be described by, for instance, Rényi, Shannon or Kolmogorov entropy measurements [48], while the interactions among its components have been described mostly by TE [49], which has been successfully used to identify leaders in interactions [50]. Mann and Garnett [51] postulated the *causal entropic principle*, according to which individuals’ behaviours in a group tend to maximize the entropy of the entire system. The causal entropic principle was indeed able to predict many social interactions in animal groups, including humans. According to these works, optimally functional biological systems should maximize the entropy values, and it follows that stressors should diminish them.

However, the reader must understand that non-linear analyses of fish behaviours are not substitutes for classical methods. Rather, they must be considered providers of valuable additional information that, as mentioned above, classical analyses do not always readily expose.

### 1.2. Aim of the Work 

By applying mathematical models and novel linear and non-linear algorithms [52,53], it is possible to extract simplified, practical information from complex biological systems (such as fish in aquaculture and in the wild), which inherently contain a large number of stochastic and deterministic components [54]. Two often used such measurements are the entropy of the system, i.e., the degree of predictability/chaoticity of the system [48,51,55], and its FD, which is a characteristic of fractal structures describing their complexity [43,44,45,56,57,58,59]. Indeed, Alados et al. [60] postulated that some exploratory biological structures and behavioural patterns have been naturally selected towards increasing complexity. Since stress increases an organism’s metabolic rate and energy consumption, it should provoke a consequent reduction in the complexity of exploratory behaviour even though humans may not be able to perceive it by eye. However, the FD of the structures and behavioural patterns, as a measure of their complexity, may serve as stress indicators and allow for the quantification of changes in behaviour. The authors demonstrated that the FD values of some behavioural complexity patterns in goats (*Sarcoptes scabieis*) (e.g., head lifts, feeding gaps, and vigilance behaviour) decreased with the two kinds of stress tested: pregnancy and parasitic infection [60].

There is also an increasing wealth of research devoted to the development of deep learning methods and neural network classification techniques using sophisticated mathematical models that will undoubtably find applications in IA and PFF (for further details, see a recent review by [7]). Many of these approaches use the entropy of the raw data (including that of images [61]) to improve the accuracy of a model. Feeding status is often selected as a targeted variable of clear interest for farmers [62,63]. As mentioned above, these applications have not been included in the present review.

The purpose of this work is to provide an overview of recent publications indicating the potential of entropy and FD analyses of fish behaviours to provide relevant information to farmers when integrated into an PFF/IA architecture, particularly with respect to the early detection of stress and critical behaviours (Figure 1). 

## 2. Methodology Followed for the Review

Initially, the Web of Science (WoS) and Scopus databases were used for the bibliographic searches, but the WoS rendered a very large number of results, most of which were not relevant for our purposes. Therefore, the searches were limited to Scopus. The search strings and number of documents found (updated on the 26 January 2023) are listed in Table 1. Reviewing all the documents, we found only 37 dealing with the exact subject of our study, namely, relevant individual or collective behaviours whose FD and/or entropy analyses have the potential to provide relevant information applicable in a PFF/IA framework. A summary of the contents of those publications is presented in Appendix A.

## 3. Targeted Applications

Appendix A displays a summary of the contents of the 37 selected publications, including the most relevant data and results. The works have been classified according to the investigated subject and the targeted applications addressing the following topics: individual identification in groups; preying/search behaviours in larvae and fish; feeding status; collective behaviour, including the effect of the number of fish, individual interactions, hierarchies, and collective behaviour in shoals of mixed species; effect of tagging and pain; fear/anxiety responses to predators; modulation of fear/anxiety and positive emotional contagion. 

### 3.1. Individual Identification in Groups

Neumeister et al. [46] were able to distinguish between the swimming trajectories of individual goldfish within groups using the discriminant analysis of six variables, consisting of the mean velocity and five nonlinear measures, namely, the characteristic FD, the Richardson dimension (D_R_), the Lempel–Ziv complexity, the Hurst exponent (HE) and the relative dispersion (the reader should refer to [46] and references to the original works describing the parameters for further information). In addition, although not apparently distinguishable, the swimming patterns were rather complex, and each fish displayed highly individual and disparate swimming profiles. No single measure was “the most effective”, although the nonlinear measures were more effective than the mean velocity, and the most effective measures were HE and D_R_.

### 3.2. Preying/Feeding Search Behaviour 

#### 3.2.1. Larvae

Coughlin et al. [64] published the first work applying fractal analysis to the swimming and preying behavioural patterns of clownfish larvae, which was conducted because traditional methods did not differentiate foraging modes appropriately. Both the age of the larvae and the amount of available prey influenced the swimming and search patterns: the larvae showed highly variable, complex swimming paths during the first two days after hatching as reflected in their higher FD, which decreased by the third day after the start of active feeding, thus denoting a trend toward simpler, more linear paths. Different search patterns were followed by the larvae after the onset of feeding: straighter (with the lowest FD) and intricate (higher FD) paths in low- and high-prey-density media, respectively, were observed. 

Mahjoub et al. [65] also studied the prey search behaviour of malabar grouper, *Epinephelus malabaricus*, larvae via the 3D recording of their swimming behaviour in the absence and presence of prey. FD analyses of the swimming projections in the three axes (XZ, XY, and YZ) revealed anisotropy in the changes of the search pattern induced by the addition of prey. Without prey, the FD analyses indicated increased activity on the vertical axis but, in the presence of prey, the complexities in the vertical and horizontal axes were similar. This indicates an optimization of the search volume by the larvae and, therefore, the need to consider 3D search behaviour in further studies. It is important to remark that the authors emphasized that the FD of the trajectories was used for comparative purposes and not to determine the exact FD of a given path. 

#### 3.2.2. Fish

The fractal properties of fish school trajectories were examined by Tikhonov et al. in two papers [44,45] using the computer modelling of four components of the trophic chain: nutrients, phytoplankton, zooplankton, and fish. Their model was able to describe a wide range of fish school motions. For example, it revealed that the groups swim to the areas with the highest zooplankton density and that the complex motion of the school is basically dependent on the predation rate. It also indicated that a decrease in the predation rate induced a transition in the type of trajectory from one with frequent changes of direction (with fractal properties for all temporal scales,) to another, straighter one (with pronounced multifractal properties for large-scale displacements). These results agree with the above-mentioned work regarding larvae after the onset of feeding [64] that displayed straighter swimming paths (with lowest FD) in a low-prey-density environment. Although not considered in their model, other works had indicated that the natural variation in environmental parameters can cause the school–prey system to adopt quasi-periodic behaviour and display chaotic oscillations ([44,45] and references therein).

Examining in the wild the prey-searching methods of tagged marine organisms from several taxa, Sims et al. [66] concluded that they adopt Lévy-like moving behaviour, albeit not continuously, to optimize their chances of encountering patches of prey. It must be noted that prey (and feed on a farm) do not usually have a random distribution. This behaviour will be particularly advantageous in environments with shifting resources due, for example, to the exploitation of their prey by other species (which may resemble an open, multitrophic aquacultural production system), climate change, or other modifications, and its study may contribute to the optimization of feeding strategies in conventional aquaculture and in PFF.

Zhang et al. [67] developed a procedure for classifying small fish groups according to their shoaling/feeding status. Processed 2D video images of zebrafish were used to create a database of different behaviours that constituted the core information used to train a VGG-16 network (a 16-layer deep convolutional neural network). Changes of the status of fish during feeding were described by two-dimensional image entropy (calculated according to SE). The procedure was able to discriminate between two group behaviours that the authors termed “Normal” when shoaling unmolested and “Abnormal” when responding to a feeding stimulus, where the response resembles the schooling of the fish. It must be noted, however, that “abnormal” is an unfortunate selection of name given that the response examined is the normal response of the fish to the feed. In any case, the authors indicate the need to develop their model further by including images of different quality and different types of behaviour. This work supports the previously mentioned study by [35] who found that not only the SE of the fish system increased with the number of fish but also that the behaviour corresponding to schooling (in response to a stochastic event, similar to the “Abnormal” behaviour described in the study by [67]) had a higher SE value than that of the “Normal” shoaling system (termed “basal” in the study by [35]), for which the actual values are a function of the number of fish in the group, a parameter whose value was not taken into consideration by [67]. For 25 fish and depending on the tank and the number of days after the beginning of the experiment, the SE in the study by [35] increased from 4.3–4.8 (basal, normal shoaling) to 4.4–5.5 (schooling, “abnormal” response), while in the study by [67] the entropy of the zebrafish system (which seems to contain over 25 individuals) in their “normal” state is recorded to be under 0.25, increasing to 0.45 upon human intervention or during feeding. 

### 3.3. Feeding Status

A relevant piece of information for farmers is the feeding status of their fish, i.e., whether they are feeding and how much they consume. While there are works devoted to the discrimination of feeding/non feeding fish, there is a lack of information on how to classify feeding status according to the amount of food a fish feeds, particularly under real-life production settings. This subject was addressed by Chen et al. ([61] where the abstract is in English and the original paper in Chinese) who developed a fine-grained classification algorithm of fish-feeding status. The raw data used to build the database were 752 videos (3 s each, constituting 90 frames) labelled as non-eating, weakly eating, or strongly eating, that had been recorded in an RAS production facility (although, unfortunately, the authors do not mention the species). An optical flow algorithm was applied to convert the videos into many inter-frame motion feature samples upon which a five-layer (comprising one input layer, one output layer, and three hidden layers) classification neural network (CNN) was built, which had three output categories corresponding to the three different selected feeding states (non-eating, weakly eating, and strongly eating). The CNN had been optimized by a cross-entropy loss function. Two issues can be raised regarding the work: One is that the CNN does not include the ability to identify (and classify) “non-feeding” fish, which is very relevant. The second is that, at least in the English summary, there is no information on whether the model was tested and/or validated with video recordings different from those used to develop the CNN. In any case, the approach is interesting and deserves further attention. 

### 3.4. Collective Behaviour 

Nonlinear approaches have helped to understand how collective decisions are better than individual ones in social groups [68], including fish. Mann and Garnett [51] successfully applied the causal entropic framework designed by [69] to determine the origin of collective behaviour from a purely entropic point of view. The authors were able to predict the fundamental form of social interactions and showed that the causal entropic principle could provide a purely statistical prediction for many of the emergent properties of collective behaviour, including cases where the mechanisms for inter-individual interactions were not fully understood. Entropy has also found applications in the detection of special behaviours of relevance for the welfare of Nile tilapia (*Oreochromis niloticus*) in a RAS system. Zhao et al. [70] detected gastrointestinal evacuation using a modified kinetic energy model (KEM) that employed the dispersion, velocity, and turning angle of the shoal as parameters, for which dispersion was estimated by optical flow, entropy, and statistical parameters. The proposed KEM model showed a good performance in detecting emergent gathering and scattering behaviours.

It is known that collective changes from shoaling to schooling in a group of fish may happen suddenly as a consequence of a stochastic event [71], and this is reflected in changes in the SE of the system [35]. This phenomenon prompted Crosato et al. [72] to study the flow of information in a school of fish during collective U-turn swimming changes by using TE. The authors identified two different flows of information: an informative flow (positive TE) from fish that have already turned to fish that are turning, and a misinformative flow (negative TE) from fish that have not turned yet to fish that are turning. Local behavioural changes of a single individual could lead to large transitions between the collective state of a school, such as between schooling and milling (an ordered state in which individuals constantly rotate around an empty core [73]). Considering that a stochastic event may be anything unexpected that will startle a fish (such as noises or the sudden appearance of a predator, feed, a diver, or a robotic camera, all of which occur under normal farming conditions), it follows that these kind of studies are of relevance when attempting to understand the dynamics of fish motion, behaviour, and communication within cages, particularly in large offshore cages. 

### 3.5. Effect of the Number of Fish

The number of fish in a group influences both the FD [74] and the SE [35] of some behaviours, particularly for small numbers of fish of shoaling species. Kushida et al. [74] calculated, among other properties, the FD of Japanese horse mackerel (*Trachurus japonicus*) from a HD video recording with a camera mounted on top of an experimental tank. Their aim was to build small virtual robots generating actions resembling the behaviour of actual fish. Their work showed that as the number of fish increased from one to twenty-four, the FD of the real fish tended to decrease, and the fish tended to move toward smaller regions of the tank. They postulated that this movement toward smaller regions should result in lower average swimming speeds that would, in turn, produce the measured lower FD values.

Unlike in the research conducted by [74], where it was shown that the FD tended to decrease with an increasing number of fish, [35] measured an increase in the SE in fish systems consisting of between one and fifty fish in both their shoaling and schooling responses for which a power relationship was maintained with the number of fish and whose coefficient of variation, which was the largest for the one-fish systems, decreased concomitantly with the number of fish. Compared to the shoaling, basal state, SE was usually higher (i.e., the unpredictability increased) for the schooling response to a stochastic event for all fish groups, but the difference was particularly noticeable for individual fish and for groups with only two to five fish. These results may indicate the influence of stress that occurs when shoaling fish find themselves isolated or in small groups, indicating that the results of experiments performed using one or a small number of fish may not be directly applicable to real-life fish-farming settings, and even less so to the larger PFF operations with hundreds of thousands of fish.

### 3.6. Collective Behaviour and Individual Interactions

Hiramatsu et al. [75] modelled the schooling behaviour of a group of medaka based on how an individual fish interacted with its neighbours in a tank with no flow, measuring the distance between individuals, their spatial distribution, and their communication processes. The structure and characteristics of the real fish school were quantified by the nearest neighbour distance (which also yielded an estimate of the cohesiveness of the school), the level of polarization (the intensity of the parallel orientation of a fish school), the expanse (an estimate of the distance of every fish to the mass centre of the school), and its FD (as a measurement of the straightness/tortuousness of the trajectory of the school’s centre of mass). The simulations were able to satisfactorily reproduce the behaviour of the medaka school using a simple genetic algorithm with fitness defined by those four variables.

The coexistence of order and flexibility within fish schools was examined by Inada and Kawachi [43] using a simple numerical model and a computer simulation. According to their results, both the number of neighbours interacting and the randomness of individual motion influence the order of the school and its flexibility. For high flexibility, there were optimal and low (two to three) numbers of interacting neighbours. A slightly larger number of interacting neighbours (four to five) was necessary when the fish paid attention to more conspecifics in the school. School order was established for three to four neighbours, indicating that schooling fish have evolved a specialized ability to establish both school order and flexibility when the maximum number of individuals interacting is relatively small, i.e., higher than three. 

Suzuki et al. [76] designed a model after quantifying the behaviour of chicken grunt (*Parapristipoma trilineatum*) schools of different sizes in a tank containing a column. The FD analyses of the time series [77] quantifying their behaviour indicated that it was random, but the attractive and repulsive forces of the walls were larger, while the magnitudes of the propulsive force were smaller, in small schools (one to five fish) than in larger schools (ten to twenty-five fish). However, the attractive and repulsive forces of the structure were significant only for the larger school (n = 25). Although the model seemed to be useful, the authors indicate the need to collect more data on different patterns of use of space and for different species. 

In a comprehensive and very interesting work, Wark et al. [78] examined the existence of population-specific shoaling behaviours in 13 wild populations of three-spined sticklebacks (*Gasterosteus aculeatus*). Standard measures of shoaling behaviour failed to distinguish among the different populations, but when their swimming patterns were analysed using population-level probability distributions and their SE was quantified, significant differences between populations were identified. Unfortunately, there were not enough data to estimate the entropy for individual fish and, consequently, it was not possible to elucidate whether the differences between populations were due to consistent individual behavioural patterns or to differences among individuals within the populations. However, a log-likelihood analysis showed that an individual’s behaviour was more akin to that of its own population, thus supporting the hypothesis that the SE of a population most likely reflects that of its individuals.

Fish are exposed to different kinds of stimuli in farms. Obviously, one of them is the feed, but there are also undesirable stimuli that may alarm fish and provoke stress and changes in their behaviour, such as noises and the presence of predators, particularly when the latter are actively feeding upon their conspecifics. Schaerf et al. [79] studied the changes in individual and group behaviour as well as the rules of interaction among fish before and after being submitted to two different stimuli: an alarm cue (the filtered, macerated bodies of their conspecifics) and a food cue. The controls were exposed to the conditioned water only. The behavioural responses were measured by linear and nonlinear parameters. Compared to the controls, the fish stimulated by food displayed a reduced frequency of observing group mates at close proximities and they travelled at higher speeds. Unsurprisingly, the strongest responses were elicited by the alarm cue, including a tendency to travel at lower speeds. Changes in their conditional entropy, mutual information, and entropy rate indicated an increase in the unpredictability of their movements upon exposure to the alarm. The alarmed fish also moved in more tightly clustered shoals with smaller distances to their neighbours than in either the control groups or the groups exposed to food. An increase in the entropy in the alarmed shoals was also reported by [35], and the difference between the SE of basal and the alarmed states was more pronounced when the fish were alone or in small groups (two to five individuals) than in larger groups of up to fifty individuals, wherein the fish may have felt safer. 

Cross-entropy loss was one of the parameters used by Heras et al. [80] to develop a deep attention network to understand, and ultimately predict, the behaviour of a fish in its group, with the group being variable in size. The example selected was the estimation of the probability of a fish within a group to turn right or left. The network was developed and trained based on the behaviour of groups of 60, 80, and 100 juvenile zebrafish that were 2D-video-recorded and whose 2D position, velocity, and acceleration values were tracked by Idtracker.ai (https://idtrackerai.readthedocs.io/, accessed on 21 March 2023). The network performed satisfactorily and, in agreement with the abovementioned works on the number of fish and their interactions, the number of interacting individuals was found to be variable, typically somewhere between 8–22 fish, with 1–10 more important cases, particularly if some moved at higher speed in front or to the sides or if they were very close or on a collision path. Their results indicate that each fish decides the turning size by actively selecting information from the group.

The work of Wilson et al. [81] presented some practical implications when a shoal of fish consists of hungry and sated fish and indicates that even optimizing the feeding conditions for all fish may decrease the expense of energy that hungry fish and groups consisting of mostly hungry fish will use to swim faster, possibly, as indicated by the authors, to optimize their detection of prey/feed. Their work examined how the number of hungry fish in groups of up to eight individuals of X-ray tetras (*Pristella maxillaris*) influences the behaviour of the individuals and the group. The authors used five different kinds of groups with various ratios of hungry/satiated fish (8/0, 6/2, 4/4, 2/6, and 0/8). They calculated, at the individual level, the mean speed and mean nearest neighbour distance and, at the group level, the median polarization (as a measure of group coherence), while information flow within each group was calculated by the mean pairwise TE, as described by [72]. Groups with a greater proportion of hungry fish swam faster and exhibited greater nearest neighbour distances, but there was no difference in the swimming speeds of hungry versus well-fed fish within the groups. Thus, the nutritional status of individuals within the group seems to impact both individual and group behaviours. In addition, one very interesting result was that the flow of information was greater in the groups with a majority of hungry individuals, but there was not a linear relationship between the pairwise TE and the relative number of hungry/fed fish. Rather, there seemed to be a difference, which was also underlined by the authors, between the three groups consisting of mostly hungry fish and the one with up to 50–50% hungry/fed fish on the one hand and the two other groups consisting of a majority of fed fish (six fed/two hungry and eight fed fish) on the other. The existence of such a breaking point, which deserves further study and characterization, may find a practical application when optimizing feeding schedules to identify the point in time when most of the fish on a farm are hungry and need to be fed.

### 3.7. Collective Behaviour and Hierarchies

Collective movement requires each individual animal to decide to move, a process that is usually initiated by a single individual or by a small group and that may be based on a variety of factors (e.g., age, sex, position in the hierarchy, etc.) [82,83]. Therefore, the identification of leaders in groups and the mechanisms of leadership are important, particularly (but not only) in the wild, where they may need to escape predators, find food, and avoid polluted and stressful locations. Collignon et al. [38] used the idTracking software [84] to generate the individual trajectories of each individual in groups of n = 2–10 zebrafish. The distribution of leadership was quantified by calculating the entropy associated with the time series of the identity of all the leaders. The results indicated that any fish had the potential to lead the collective movement of the group and that all shoal members had the same success rate. The predictor of a fish’s tendency to initiate collective movement mainly seemed to be its mobility, regardless of its position in the hierarchy, i.e., an informed individual moving in a specific direction is more likely to be followed by a group of naive individuals by just moving faster than the rest. However, it was also observed that some individuals lead collective movements more often than others, which may, over time, result in the development of specialized roles, for example, if this initiation is linked to personality traits such as boldness.

Niizato et al. [85] used ayus (*Plecoglossus altivelis*) and boids (artificial life simulation) models to assess the integrity in small groups of fish (n = 2 to 5). They compared three parameters to analyse the dynamics within the fish groups: Mutual information (MI), TE, and integrated information theory (IIT 3.0). Unlike most parameters concerning the acquirement of information about “what the (fish) system does”, integrated information theory (IIT) measures the “degree of consciousness” of a system. IIT postulates that the amount of information loss caused by a minimum information partition is equivalent to the degree of information integration in the system, i.e., IIT measures “what the (fish) system is” [85]. IIT 3.0 identifies intrinsic differences in the integrity of schools of n = 2–5 fish, and it detects the existence of a discontinuity in the integrity of the system between groups of n = 2–3 and those n = 4–5 individuals. Such differences in group integrity (i.e., an estimation of leadership in the group, albeit different from the classical leadership concept as used in TE analyses) is due to the fact that group integration in systems of n < 4 fish requires an intact visual field, but for groups of n > 4 fish, group integration has tolerance for some blind spots. No such discontinuity was detected by the SE of seabass groups of n = 1 to 5 fish mentioned above by [35]. This may be due to differences in the parameters used but also to the time scale of the analyses (3 min in the study by [35] and 0.05 s in the study by [85]) and to the likely differences in the behaviour of the two species. It is important to stress that the results obtained by [85] are only applicable within the very limited timescale used (the reaction time of a fish, which is about 0.05 s) and to the number of fish used (n = 2–5). The behaviour of systems over longer time periods and incorporating a larger number of fish must be examined independently. Nevertheless, IIT has emerged as an interesting parameter with respect to understanding fish systems’ behaviour and may shed new light on the understanding of the behaviour of larger systems, such as those used in PFF. 

### 3.8. Collective Behaviour and Mixed Shoal Species

Despite the emphasis placed on multitrophic aquaculture, there are very few published works on fish behaviour in mixed-species settings. It must be noted that the species mixed on multitrophic farms do not predate on each other; thus, the well-studied predator–prey interactions should not apply to those cases. Ward et al. [86] studied the behaviour of three species that form mixed-species shoals in the wild: three-spined sticklebacks (*Gasterosteus aculeatus*), nine-spined sticklebacks (*Pungitius pungitius*), and roach (*Rutilus rutilus*). The authors set up different experiments in which the individuals from the three species were mixed and analysed their behaviour through classical (the mean of the median speeds of each fish, mean distance between all fish, and the mean polarization of the group during each trial) and nonlinear (TE) methods. The single-species groups were more polarized than the mixed-species groups, and there were differences between treatments in terms of the mean pairwise TE. Species-specific differences were noted with respect to (1) the use of information within the mixed-species groups and (2) the responses to conspecifics and heterospecifics in the mixed-species groups. The TE-based results indicated that information flows both between and within species in all treatments. Interestingly, the results from the TE analysis did not always agree with the results of the classical parameters uncovering features of the inter-species interactions that traditional measurement methods did not. This study stresses the relevance of examining the inter-species interactions in multitrophic settings and the need to use non-linear methods to monitor their interactions. 

### 3.9. Tagging and Pain

The tagging of fish is a common practice used to perform experiments and, potentially, in farming settings to identify fish and monitor their production [6]. The Visual Implant Elastomer (VIE) tags developed by Northwest Marine Technology are considered to avoid significantly influencing the behaviour of fish [87]. Corroborating the work of [87], Eguiraun et al. [23] also found a negligible effect of VIE-tagging on the SE of the schooling response of a group of 81 European seabass (a 0.25% decrease) and a very small effect on the permutation entropy of the same response (increased by less than 1%). On the other hand, the Katz–Castiglioni FD of the same experiment [23] decreased in tagged fish somewhere between 4–15% (for sliding window lengths of between 320–1280 points from 24 frames/sec video images). It must be noted that the Katz–Castiglioni FD suffered an increase upon MeHg treatment between 3–15%, which led the authors to propose a non-relevant effect of the tag compared to that of a documented chemical stressor [23].

Using zebrafish, Ruberto et al. [88] found somewhat different results: tagging barely impacted individual behaviour and shoaling and schooling tendencies, but the procedure did increase the speed of the individual fish and that of the group. Interestingly, TE indicated a significant effect of tagging on inter-individual interactions, providing indications that they might be tag-colour-dependent. For example, yellow-tagged fish were less likely to influence others, although the other colours did not seem to provoke similar responses [88]. The influence of tagging on the social behaviour of fish had already been documented by Frommen et al. [89], who showed that test fish spent significantly more time near the tagged shoal than near the sham-tagged shoal; however, contrary to [88], they did not find any significant effect attributable to the colour of the tag. However, in the study by [89], the test zebrafish that were not tagged and had grown with non-tagged fish did show a preference to swim closer to tagged individuals. Taken together, it seems that tagging may have a colour- and species-specific effect on fish behaviour, and caution should be adopted when experiments require the mixing of tagged and non-tagged fish.

Tagging most likely also causes pain, which inherently brings about reduced welfare and increased stress. While VIE tags are expected to induce some form of pain in a fish when injected, this tagging procedure is usually performed under anaesthesia, which may exert an effect on its own (see below), but other forms of tagging may be performed quickly and without aesthetic treatment, such as fin clipping. When the tagging and/or the tag induce pain, the pain itself may change the behaviour of the injected fish, as demonstrated by Deakin et al. [90]. These authors examined the effects of two alternative tagging methods, namely, fin clipping (performed with and without lidocaine anaesthesia) and PIT tagging (the implantation of Passive Integrated Transponders for individual fish identification). They assessed the effects of the treatments on the complexity of individual fishes’ behaviour by measuring the FD of their 3D swimming trajectories and showed that, indeed, both treatments decreased the complexity of their behaviours, which was reflected by a decrease in their FD values. Interestingly, an injection of acetic acid, a standard pain test, also decreased the FD in a dose-dependent manner. On the other hand, when the fins were clipped after lidocaine treatment, the FD of the fish was not altered, resembling that of the control and sham-treated fish. This work indicates the usefulness of the FD of swimming trajectories for estimating changes in fish welfare due to pain and the beneficial effects of the use of anaesthesia to avoid painful procedures.

Fin amputation and pain in zebrafish were also the subject of a very interesting study by Audira et al. [91]. Fin tagging is a common procedure used to distinguish fish groups under experimental conditions where different fins may be targeted for different experimental groups in the same experiment. The study also addressed whether the selection of the fin (dorsal, caudal, anal, pelvic, or pectoral) might have implications on the behaviour of the amputated fish. The authors aimed to assess the complexity of 3D swimming behaviour (by idTracker) by measuring the subjects’ locomotor activity (by four end-point parameters), movement orientation (two end points), exploratory behaviour (six end points) and through FD and entropy analyses of their swimming trajectories. Additional experiments were performed to verify whether potential changes might be attributed to pain (by using lidocaine) or whether they were simply due to mechanical effects caused by the lack of a fin. The subjects’ behaviour during the regeneration period was also examined. The observations during the first 2 days after fin amputation were made for isolated fish and groups (n = 6). Upon identification of caudal amputation as the intervention exerting the strongest (negative) effect on fish behaviour, the caudal-fin-amputated fish were further investigated, but only in groups (n = 6), for 10 days for the lidocaine-treated fish and a further 30 days to perform follow-up assessments of their behaviour during caudal fin regeneration. The FD values of fish behaviour for the fish with amputated caudal, pelvic, or pectoral fins were significantly lower than those of the control fish but only when the fish were tested in groups and not when they were tested individually. Similarly, only the entropy of the fish groups whose caudal fin (but not other fins) had been amputated was significantly different, albeit higher, than the entropy of the control groups, but the significance was lost when the fish were individually tested. According to the behavioural parameters, recovery started 1 day post-amputation, was almost complete after 5 days, and was fully completed by the 10th day. It was paralleled by a gradual recovery in the values of FD and entropy, which, 3 days post-amputation, resembled the controls. The regeneration of the caudal fin had already started on the 5th day, while almost full and full regeneration had occurred by days 20–25 and day 30 day, respectively. However, only 5 days post-amputation, with only about 20% of the fin regenerated, the fish were able to display relatively normal behaviour, indicating that mechanical hindrance may not be the main reason for their altered behaviours. Lidocaine treatment provoked diminished activity, which is an expected side-effect consistent with its sedative effects. The diminished activity of amputated fish may also be a response adopted to avoid the aggressive responses of conspecifics known to occur in species that form dominance hierarches, such as zebrafish, and whose intensity may be dependent on the size of the group (see the research conducted by [91] for a complete discussion and additional references). As the authors indicate, this would explain why the fish tested individually (i.e., in the absence of threatening conspecifics) displayed higher activity.

Although the study by [91] was performed on zebrafish and to test pain, it casts some very interesting light on the application of entropy and FD techniques for identifying the behaviour of farmed fish that have been subjected to aggression from conspecifics, given that such aggression is often enacted in the form of bites to the fins. Moreover, it can help to quantify how practical implementations to reduce stress, e.g., through environmental enrichment, may decrease aggression [92,93]. It may also explain why wounded fish would prefer to be still and alone (thereby avoiding potential aggression) and why healthy fish might prefer to be in groups (safety in numbers).

### 3.10. Fear/Anxiety Responses to Predators

Regarding the study of fishes’ responses to predators, robots have been introduced in research studies as predators in order to prevent focal fish from exhibiting the inconsistent responses that are often observed when real predators are introduced in the system. This is because unlike actual predatory fish, robots can be used under controlled, custom-designed experimental conditions that produce more repeatable results.

Butail et al. [94] used zebrafish in an experimental set up with a known information flow and applied TE as a measure of directional information in two systems: (1) fish–fish and (2) fish–robot (a life-sized zebrafish replica) interactions. The behaviours in the two systems were 2D-video-recorded by a camera above the tank, and the SE was used to calculate the TE of the interactions between the fish–conspecific and fish–robot. As expected, the authors found that when a live fish interacts with a replica moving along a predetermined trajectory, the dominant flow of information is from the replica to the fish, and the information flow is reduced significantly if the motion of the replica is randomly delayed. On the other hand, the TE from a living focal fish to a living conspecific and vice versa were not significantly different [94]. In a related experiment, Bartolini et al. [95] used robotic replicas of different sizes and found that the TE indicated preferences and an adjustment in fish behaviour consisting of an avoidance of larger replicas and attraction towards smaller ones, by which the fish were, in turn, influenced. In contrast to the study conducted by [94], Bartolini et al. [95] found that similar-sized replicas did not elicit significant responses, i.e., no information transfer was observed between the fish and surrogate data generated by the hypothetical motion of either a support or a shoal of replicas.

TE was also applied by Neri et al. [96] to identify causal relationships in the behaviour of fish using a prey–predator system where the predatory fish was located outside the space occupied by the prey. This set up imitated the conditions of a fish swimming in a cage and visually interacting with predators attracted to the net and surrounding it. They initially used a robotic predator and, as is common when one of the interacting partners is a robot, the information flowed unidirectionally from the robot (predator) to the fish, i.e., the zebrafish reacted to the behaviour of the robot. However, when the robot was replaced by a real predatory fish (the red tiger oscar fish, *Astronotus ocellatus*), the one-directional information flow was substituted by a reciprocal one: upon visual interaction, a positive feedback loop was established by which the predator watches and shows an increased responsiveness to the prey’s movement, whose response is avoidance. This can be of relevance for fish swimming in the outer parts of the net where predators may also swim looking for potential prey. Interestingly, these results contrast with those related by Hu et al. [97], who reported a net information flow from a living predator (the northern snakehead, *Channa argus*) to the prey fish (rosy bitterling, *Rhodeus ocellatus*) in a similar circular arena. However, Hu’s work had several methodological differences, including the use of different species and the fact that each fish could feel the ripples in the water produced when the other fish swam. In this case, the flow of information moved from from the predator to the prey, including a critical and sensitive region where the prey is highly vigilant of the predator’s behaviour and did not allow it to draw near.

To evaluate a zebrafish’s fear response to a 3D replica of the above-mentioned allopatric predator (*Astronotus ocellatus*), Spinello et al. [98] used geotaxis and two avoidance-related parameters: the average distance between the replica and the fish and the time spent by the focal fish in the half of the water column opposite to that occupied by the replica. A finite-state Markov chain switching between “stationary”, “swimming”, and “attacking” states was used to control the motion of the replica. As expected, exposure to the replica increased the fear response of the fish, shown by an increased rate of geotaxis (i.e., the fish swam to the bottom), and they spent different times in the vertical axis opposite to that occupied by the replica, but there were no differences regarding the average distance from the replica. TE revealed a rapid adjustment of the fishes’ behaviour to avoid the predator’s attacks.

### 3.11. Modulation of Fear/Anxiety Responses

The parameter most used as an indicator of fear/anxiety is geotaxis, i.e., the tendency to swim to the bottom of a tank. To characterize and model this behaviour, Burbano and Porfiri [99] used data from zebrafish treated with citalopram and ethanol [100], two commonly used anxiolytic drugs, and what they defined as spatial entropy (calculated as its SE) to identify the extent of the volume of water occupied by the fish and in which part of the tank its activity is more concentrated. The same group of authors have extensively used zebrafish models to study the effect of fear/anxiety and anxiolytic drugs on this species’ behaviour and on the positive emotional contagion conferred by the treated toward the untreated fish to relieve the anxiety responses provoked in the latter by a predator (see references below).

Examining the effect of psychoactive substances on zebrafish behaviour, Ladu et al. [101] applied TE to quantify the interaction between individual fish and a shoal-replica of four zebrafish treated with different doses of caffeine (0 (control), 5, 25, and 50 mg/L). TE did not detect any significant flow of information from fish to replica and vice versa for either the control or the group with the lowest caffeine dose; however, for the fish treated with 25 and 50 mg/L, the TE from fish to replica was lower than that from the replica to the fish.

Macrì et al. [102] documented the modulation of fear conditioning in zebrafish provoked by ethanol (a 0% control group, and two test groups with 0.25% and 1.00% concentrations of ethanol/water). Individual 3D swimming trajectories of the fish were generated by two video cameras (placed on top and in front of the experimental tank) and analysed using the authors’ own software. The fear response was elicited by three zebrafish replicas manoeuvred along 3D trajectories by a robotic platform. The following parameters were tested: the avoidance index, geotaxis, freezing, spatial entropy (tendency of the fish to explore the tank, which was computed as the SE), and average speed, acceleration and angular speed. The spatial avoidance values indicated that ethanol, particularly the 0.25% concentration, decreased the degree of aversion to the fear-related compartment. Geotaxis was also modulated by low ethanol concentrations: the 0.25%-ethanol-treated fish displayed no preference, while both the control and 1.00%-ethanol-treated fish preferred the bottom of the tank. Over the course of the experiment, the evolution of the values of spatial entropy and average speed in the control and 0.25%-ethanol-treated fish were comparable, but those of the 1.00%-treated individuals decreased with time. Thus, according to the authors’ hypothesis, the zebrafish were indeed fear-conditioned, and the response was modulated by ethanol, particularly for anxiety-related behaviours (i.e., spatial avoidance and geotaxis). However, only the highest dose (1.00%) exerted effects on the other parameters (i.e., average speed, average acceleration, and spatial entropy), yet these were small and mostly consisted of a decrease in the values during the test. These results confirmed the ethanol-dependent reduction in general locomotion observed in previous studies ([102] and references therein).

In a related study, Clément et al. [103] 3D-tracked zebrafish behaviour to estimate fear-conditioned behaviour and how it is affected by the treatment with two anxiolytic substances: citalopram and ethanol. As described in the study by [104], fear was elicited by a programmed robot simulating a zebrafish-sympatric predator (an Indian pond heron, *Ardeola grayii*), which hit the water surface of a lateral compartment approximately every 30 s to induce, as a conditioned response, the avoidance of the upper part of the water column. The success of the conditioning procedure was indicated by the values of spatial entropy (calculated using the SE). The experiment was performed in the presence and absence of three different concentrations of citalopram (30, 50, and 100 mg/L) and ethanol (0.25%, 0.50%, and 1.00%) and the drug-free control groups. The parameters measured were the avoidance index, geotaxis, freezing, spatial entropy, and average speed, acceleration and angular speed. The results of the spatial entropy values for citalopram and ethanol treatments were also similar and mirrored the freezing behaviour: the fish displayed an increasing tendency to explore during the later stages of the test that was not modulated by citalopram concentration, while the spatial entropy did not significantly vary according to the experimental group, and no time–concentration interactions were found. Geotaxis was the most informative parameter: a conditioned control fish showed a clear positional avoidance of the robot by exhibiting a robust preference for the lower portion of the tank. As hypothesized by the authors, the citalopram administration resulted in a linear dose–response curve with respect to anxiety, with the subjects treated with a 100 mg/L dose exhibiting a significant preference for the upper part of the tank. Similar results were obtained with the ethanol treatment but, in this case, the modulation followed a U-shaped dose–response curve: the fish treated with 0.25% ethanol preferred the upper part, while the fish treated with 0%, 0.50%, and 1.00% preferred the lower part of the tank.

### 3.12. Psychoactive Drugs for Anxiety Modulation

Macrì et al. [100] compared the behavioural information obtained from 2D and 3D video-tracking procedures. The subjects of the study were groups of zebrafish (n = 16, consisting of 8 males and 8 females), and the treatments were as follows: a control, three groups treated with citalopram (30, 50, and 100 mg/L), and three groups treated with ethanol (0.25%, 0.50%, and 1.00% ethanol/water vol/vol). One objective of the study was to assess the behavioural responses of the fish to the drugs, for which the authors found that both drugs influenced the fishes’ swimming patterns and anxiety-related profiles. In addition, the ethanol administration induced erratic movements, freezing, and the avoidance of the anxiety-eliciting areas. The second purpose of their work was to compare the quality of the information obtained from the 2D and from 3D video recordings. The results indicate that compared to the 3D views, the 2D views occasionally yielded false positive and false negative findings. Unsurprisingly, the 2D projections of 3D trajectories introduced a source of unwanted variation in zebrafish behavioural phenotyping and both 2D views underestimated the absolute levels of general locomotion. Given a choice regarding the positioning of a camera for a 2D recording only, the authors stated that the top of the experimental tank would be preferable, since the data thus obtained were more akin to those from the 3D reconstruction. On the other hand, they did not recommend to position the camera for a frontal view, since this produced negative findings [100]. Although the work does not use entropy or FD data treatment (and, therefore, is not listed in Appendix A), we consider that the subject is relevant both as a basis with which to recommend 3D image acquisition and to understand the other related works by the same group of authors mentioned in this review that do use non-linear methods.

### 3.13. Positive Emotional Contagion

Emotional contagion, a behaviour widely documented in numerous species, including fish, seems to be unconscious and governed by the amygdala [105]. Therefore, species with a proper amygdala or with a homologous organ (such as zebrafish) may respond in a similar manner (see the research conducted by [106] for a more detailed discussion). Until now, fear-contagion-based studies have mostly been the object of scientific work, but some recent studies (mentioned below) indicate that positive emotional contagion also occurs in fish groups. The study and implementation of positive emotional contagion in fish farming may improve the welfare of the entire production by adequately treating only a subset of fish. Alternatively, the selection and inclusion of “positive” fish, i.e., those with an innate tendency to display positive behaviours, may have a similar effect on the shoal, and it would eliminate the need to use undesirable and time-dependent treatments.

Burbano Lombana et al. [106] examined the response to citalopram of individually treated zebrafish both individually (where the fish were tested in isolation) and in groups (consisting of one treated and four untreated fish). The experimental set up was akin to that described in the study by [100], and the treatment consisted of a control, untreated group, and two groups treated with 30 and 100 mg citalopram/L. In agreement with previous studies, the citalopram treatment decreased the geotaxis of the treated fish when individually tested. Interestingly, the presence of a single treated fish in the group also decreased the anxiety-related behaviour of the entire group, which was demonstrated by their reduced geotaxis. Notably, while the behaviours of the fish treated with 30 and 100 mg/L were not significantly different from each other, the fish treated with the higher dose displayed a greater tendency to swim upwards when compared to the control. Group cohesion (nearest neighbour distance) and coordination (polarization) were not affected by the treatment. Interestingly, a TE analysis of the causal interactions within the group showed that the directionality of the emotional contagion was from the treated to the untreated fish, but not the other way around.

## 4. Prospects and Research Requirements

Most of the studies reviewed herein were performed under experimental conditions and on model fish systems (primarily zebrafish), with excellent water quality and light conditions and no external disturbances. Furthermore, the information in these studies was usually obtained through the processing of 2D video images (which [100] reported to be suboptimal) and the recording of only a few minutes of activity at most, even though the videos may have been recorded for longer times and the experiments themselves may have lasted many days. Therefore, even though the results are undoubtfully relevant, they must be confirmed under real farming conditions. One of the key issues is the acquisition of high-quality reliable data, preferably from information obtained from 3D swimming patterns [100]. In our opinion, sensors will be optimized and deployed in the near future to provide 3D information on fish behaviour, regardless of the quality of water. A second key issue is the need for complete, reliable databases of recorded and analysed behaviours from different fish species, under different conditions, and concerning longer periods of time. The addressal of this issue will enable the fast and successful classification of on-farm detected behaviours through the use of AI procedures such as 3D machine vision and machine learning for the adequate classification of information (including classification convolutional and neural networks, Bayesian networks, Hidden Markov models, and others). Thirdly, the on-farm information needs to be recorded with high frequency, promptly processed, and made available to farmers given that such information should aim to both facilitate the control of production and the detection, as quickly as possible, of deviations affecting the health and welfare of fish and the quality of production. It follows that implementation of a full IA system implies a large investment, both financially and in terms of the recruitment of expert personnel, that only powerful farmers are likely to be able to afford. Yet, their willingness to share valuable information in a competitive market will likely be very limited. Fortunately, some punctual applications are indeed finding their way into practical implementation, for instance, machine learning algorithms and techniques for identifying and classifying fish, evaluating biomass, performing behavioural analysis, and predicting water quality parameters [15].

A particularly interesting and novel study examined herein is the one addressing positive emotional contagion [106] since it opens the possibility of improving the welfare of the shoal by manipulating the status of selected individuals. It also opens interesting new research avenues concerning the identification and possible selection of fish strains with naturally low levels of anxiety to exert a positive effect on the welfare of the entire production.

In our opinion, and based on the above-mentioned studies and how the field is developing, it is clear that non-linear analyses, including entropy and fractal analyses of fish behaviour on farms, will find wide applications in (1) identifying the status of individual fish in a shoal; identifying (2) normal behaviours (in response to normal environmental variations, feed, and conspecifics) and (3) abnormal ones (in response to disease, parasites, aggression, and predators); (4) quantifying welfare; (5) maintaining health; (6) reducing disease and parasitism (potentially uncovered for instance by changes in their FD); and (7) reducing stress (positive emotional contagion) and, consequently, (8) improving the yield and quality of production. However, as already mentioned above, the practical application and successful implementation of these techniques still requires a wealth of additional experimental and practical data obtained in real farming settings, under different conditions, and for different species.

## Figures and Tables

**Figure 1 entropy-25-00559-f001:**
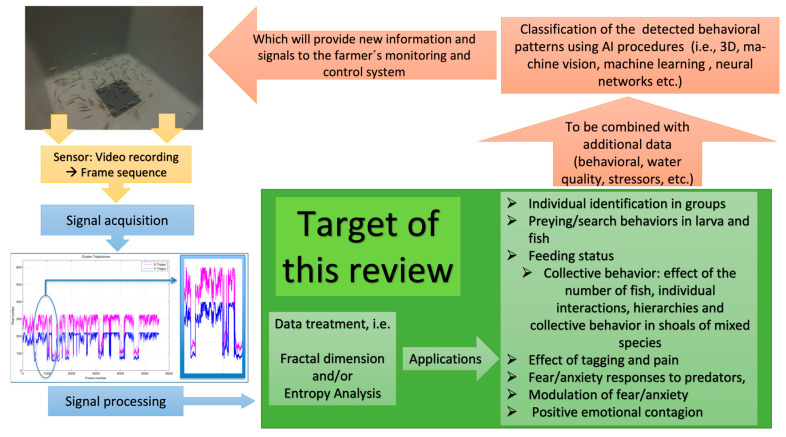
Schematic diagram indicating the subjects targeted by the review and how they would fit into a PFF framework.

**Table 1 entropy-25-00559-t001:** Search strings used in the Scopus database and number of results obtained on 26 January 2023.

Queries	Documents
(TITLE-ABS-KEY (fish AND behav*) AND TITLE-ABS-KEY (fractal* OR entropy))	143
(TITLE-ABS-KEY (aquacult*) AND TITLE-ABS-KEY (fractal* OR entropy))	87
(TITLE-ABS-KEY ("Fish behavio*") AND TITLE-ABS-KEY (entropy))	11
(TITLE-ABS-KEY ("Fish behavio*") AND TITLE-ABS-KEY (fractal))	9
(TITLE-ABS-KEY ("collective behaviour" OR "collective behavior") AND TITLE-ABS-KEY (fish) AND TITLE-ABS-KEY (welfare OR stress* OR health OR disease))	23

## Data Availability

No new data were created in the preparation of this review. All publications cited are public.

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
