# Peer review of "Entropy and Fractal Techniques for Monitoring Fish Behaviour and Welfare in Aquacultural Precision Fish Farming—A Review"

_entropy, 2023, doi:10.3390/e25040559_

Round 1

Reviewer 1 Report

Harkaitz Eguiraun and Iciar Martinez review the entropy and fractal techniques to monitor fish behaviour and welfare in aquaculture's precision fish farming. This paper shows the authors' deep understanding of this field and provided useful information. It is a very well-written paper. I suggest this paper be published after several modifications. Below are my suggestions.

Major suggestions:

(1) If I am correct, this paper only had one table and zero figures. I suggest the authors draw one "TOC-ART" related figure and help the readers to understand the papers (e.g., structure and significance) better. 

(2) Abstract directly started with a very professional sentence like "Non-linear dynamical analyses of biological systems..." It may be difficult for some readers. I suggested the authors provide one or two sentences of "general knowledge" at the beginning and then transition to the "non-linear dynamical analyses".

(3) I don't have the word count and I think it is wordy in some long paragraphs. However, the authors can reduce the number of sentences and provide more concise writing on key information. It will help the readers to learn the key points.

Minor suggestions:

(1) Line 116 The authors write [[28]. Please correct.

(2) Line 321 "Effectof" should be "Effect of".

(3) Line 689 "ethanol (0.25, 0.50, and 1.00%)" The authors have only one %... Line 703 " 0%, 0.50% and 1.00%" & Line 708 "0.25%, 0.50%, 1.00% ethanol/water vol/vol" The authors have % for every number.

Examine all other "number + units" places and make writing consistent before publication.

(4) Line 737 The authors write two [104][104] in one position. Please revise.

(5) Line 760-766 One emerging and popular trend is using a lot of images by camera trap and models by a convolutional neural network. This technique may be very important. The authors should carefully discuss future prospects.

Author Response

Reviewer #1

Comments and Suggestions for Authors:

Harkaitz Eguiraun and Iciar Martinez review the entropy and fractal techniques to monitor fish behaviour and welfare in aquaculture's precision fish farming. This paper shows the authors' deep understanding of this field and provided useful information. It is a very well-written paper. I suggest this paper be published after several modifications.

We thank the reviewer for the comments. The suggestions for modifications have been addressed as follows:

Major suggestions:

Q1: If I am correct, this paper only had one table and zero figures. I suggest the authors draw one "TOC-ART" related figure and help the readers to understand the papers (e.g., structure and significance) better.

A1: The referee is correct the paper only has one table summarizing the results of the reviewed papers and no figures. Following the indications, we have added a figure to make the manuscript more attractive indicating its structure and the subjects targeted by the review.

Q2: Abstract directly started with a very professional sentence like "Non-linear dynamical analyses of biological systems..." It may be difficult for some readers. I suggested the authors provide one or two sentences of "general knowledge" at the beginning and then transition to the "non-linear dynamical analyses".

A2: To comply with the suggestion, we have added the following 2 sentences in lines 15-17: ”In a non-linear system, such a biological system, the change of the output (f.e behaviour) is not proportional to the change of the input (f.e. exposure to stressors). In addition, biological systems also change over time, i.e. they are dynamical. Moreover…”

Q3: I don't have the word count and I think it is wordy in some long paragraphs. However, the authors can reduce the number of sentences and provide more concise writing on key information. It will help the readers to learn the key points.

A3: Without a more concise indication of which paragraphs the referee finds too long it is difficult for us to satisfy this suggestion. We have corrected some errors and edited the manuscript, particularly shortening some sentences, to facilitate its reading. However, we had already condensed the relevant information contained in the papers reviewed as much as possible and we are not able to find superfluous information. In any case, what the reader might consider key information and find superfluous will depend on his/her background and we do not think that shortening it will improve its quality.

Minor suggestions:

Q4: Line 116 The authors write [[28]. Please correct.

A4: It has been corrected.

Q5: Line 321 "Effectof" should be "Effect of".

A5: It has been corrected.

Q6: Line 689 "ethanol (0.25, 0.50, and 1.00%)" The authors have only one %... Line 703 " 0%, 0.50% and 1.00%" & Line 708 "0.25%, 0.50%, 1.00% ethanol/water vol/vol" The authors have % for every number. Examine all other "number + units" places and make writing consistent before publication.

A6: We apologize for this. All the number+units have been made consistent all along the revised manuscript with the units only once at the end of the values, ie.: 0.25, 0.50, and 1.00 %

Q7: Line 737 The authors write two [104][104] in one position. Please revise.

A7: One of them has been deleted.

Q8: Line 760-766 One emerging and popular trend is using a lot of images by camera trap and models by a convolutional neural network. This technique may be very important. The authors should carefully discuss future prospects.

A8: The reviewer is correct and indeed those approaches do deserve serious consideration. The original version already mentions this approach (see lines 96-100) but a “careful discussion” of this and related technologies, all of which have the potential to be very important for classification purposes, would not be appropriate in this work. As we hope Figure 1 will help to clarify, this review does not include discussions on signal acquisition methods (and here one could discuss whether “a lot” of 2D images might be optimal when 3D information perhaps should be given priority) or information classification techniques (of which, as mentioned CNN is only one of them). We do cite experimental and review papers the reader should refer to for additional information. We have modified a sentence in this last section to specifically add “convolutional neural networks” as a technique deserving further consideration, in lines 314 and 863-865.

Reviewer 2 Report

Title: This article summarizes recent works on the application of non-linear dynamic analysis to fish collective behaviour and proposed such methodologies to improve welfare and food quality production in fish farming. However, the title does not mention the type of article. I suggested including "review" or any equal term that identifies this work as a review article

Line 20: "predict its evolution". I would suggest removing “evolution” of non-biological event, i.e. fish farming industry. A better choice would be “improve” or similar term.

Lines 22-23: I would suggest explaining the terms “emotional contagion”. All other behaviours and conditions are well identifiable and self-explained, while “emotional contagion” required some more description.

Lines 39-40: I would suggest separating the fishing industry into two macro-categories: land and sea farms, which are more subject to climate change, and those in greenhouses or covered structures, subject instead to problems of overcrowding and other negative conditions that lead to stress. Furthermore, I think it is necessary to specify this since the acquisition of good images takes place easily in the secondary type of structure compared to those at sea, where it is not yet possible or with many difficulties to be able to do so (i.e., sunlight, electronic issue, turbidity of water).

Line 77-78: It would be appropriate to quote other reviews and expanded this topic.

Lines 82-83: There are works where using CNN that can categorize behavioural patterns, such as foraging or aggressive interaction, and therefore already applicable or in implementation perspective to evaluate these behaviours (e.g., Yang, X., Zhang, S., Liu, J., Gao, Q., Dong, S., & Zhou, C. (2021). Deep learning for smart fish farming: applications, opportunities and challenges. Reviews in Aquaculture, 13(1), 66-90.). Therefore, authors must would discuss the advantages and disadvantages of such techniques and motivate how a non-linear dynamic approach could overcome this limitation.

Lines 95-97 and paragraph 1.1:

I have two major points that authors should be considered.

For what my concern, it is missing an explanation of the reason why social hierarchy negatively influence individuals. In natural population, social structures are present and widely share among species because few individuals control resources. However, these structures are dynamic and change over time within and between generations, consequently populations have evolved and adapted to such conditions. For example, bigger individual control food resources but allowed smaller individual to exploit them. Thus, social structures per se do not cause any form of pain, at least when external condition are dynamically changing. One of the major limitation in aquaculture facilities is the predictability of resources that accentuate such aggressive behaviours and maintain them over time, thus few fish access to the resource and can grow while many others do not. By intervening on this aspect, farming animals’ conditions have seen an improvement on their survival rate and, in general, welfare. Studying the hierarchies within the groups in farms can help to understand the conditions of the group (see previous comment), but there are other aspects to consider that require relatively little effort.

There is no ecological motivation on the importance of studying these dynamics and how they can lead to improvements, from a morphological, behavioural and cognitive point of view, of farmed fish (this is described briefly in the scope, but I would expand the discussion here). Authors should motivate this aspect.

The second point concerns the algorithm. Some software acquire information on two spatial dimensions, from which different types of behaviour can be extracted, including social behaviour, validated in numerous studies (e.g., Ethovision). In addition, supplementary analysis as factor analysis (PCA) of individually collected behaviours can provide an overview of individual and social behaviour. I would suggest adding a discussion on this point and what advantages it brings to use non-linear dynamics.

Table 1: “school behaviour” has been widely used in respect to collective behaviour.

In addition, several works presented concern studies on single individuals (see “pain” section). Therefore, the review did not focus solely and exclusively on collective behaviour, and this aspect should be clarified.

Row 321: 3.5 Effect of the number of fish

Future perspectives and research needs: I would add a final paragraph on how these methodologies can be useful to producers, how they can learn and learn them, or what systems can be implemented for them to use (not all are experts and work with computations).

Author Response

Reviewer #2

We thank the reviewer for very interesting comments. The suggestions for modifications have been addressed as follows:

Comments and Suggestions for Authors

Q1: Title: This article summarizes recent works on the application of non-linear dynamic analysis to fish collective behaviour and proposed such methodologies to improve welfare and food quality production in fish farming. However, the title does not mention the type of article. I suggested including "review" or any equal term that identifies this work as a review article.

A1: The original title did not mention the word “review” because right on top of it, it is mentioned that this is a “Review Paper”. In any case, to comply with the comment, the revised title: “Entropy and Fractal Techniques to Monitor Fish Behavior and Welfare in Aquaculture's Precision Fish Farming - A review” includes the word “review”. In addition, we have also eliminated the word “paper” and used instead “review” in line 30.

Q2:Line 20: "predict its evolution". I would suggest removing “evolution” of non-biological event, i.e. fish farming industry. A better choice would be “improve” or similar term.

A2: The word evolution in line 23 does refer to the biological system (I.e., the farmed fish) as stated in the sentence “…understand the status of farmed fish and predict its evolution.” To use the term “improve” or a synonym would not be correct because the techniques examined in the review cannot improve a biological system, only document its current status and, based on additional data (hence the need for comprehensive databases as stated in the last section) and experience, help to predict its evolution. The use of the term improvement of the system’s evolution would in addition improperly imply (1) that the system is not functioning optimally, which may or not be the case (i.e. the nonlinear monitoring and prediction of the system’s evolution may indicate that it is within the optimal operational points) and (2) that there is a way to in fact improve its evolution, and that is not straightforward either, since nowadays a great deal of the knowledge necessary to improve the system’s (the fishes’) performance is either lacking or resting with the farmer’s empirical knowledge. To avoid the misinterpretation of the referee, we have rephrased the sentence to “understand and predict the evolution of the status of farmed fish” in lines 22-23 of the revised version.

Q3: Lines 22-23: I would suggest explaining the terms “emotional contagion”. All other behaviours and conditions are well identifiable and self-explained, while “emotional contagion” required some more description.

A3: “Emotional contagion” means, literally, the contagion of emotions, which we considered was also self-explained. The revised version includes the definition of “positive emotional contagion”: we have added (social contagion of positive emotions)” after the term in line 26 of the revised version.

Q4: Lines 39-40: I would suggest separating the fishing industry into two macro-categories: land and sea farms, which are more subject to climate change, and those in greenhouses or covered structures, subject instead to problems of overcrowding and other negative conditions that lead to stress.

A4: The sentence in lines 39-40 of the original manuscript refer to challenges the fish farming industry in general faces, as it is indicated in the cited document (the strategic research and innovation agenda of the European Technology and Innovation Platform). This has been indicated in the revised version (line 42 of the revised version) by adding “in general”. In any case, and it is our opinion, climate changes will heavily influence all types of fish farming settings both open and closed (mentioned by the reviewer) albeit due to different reasons. For example, closed RAS and even more so flow-through systems will likely suffer changes in the availability of clean water and being more energy consuming than open systems. Other challenges, such as the needs to ensure fish health and welfare, to develop alternative disease treatments and to identify novel sources for feeds and nutrients will affect equally both open and closed systems. On the hand, overcrowding and optimization of O2, salinity (if applicable) and elimination of metabolites are problems that farmers must address in both systems (in open cage systems for example there is a need to avoid clogging of nets). Also, fish farming cannot be divided only into the two categories mentioned by the reviewer (we believe he/she means open vs closed systems), since it is possible to apply other criteria that are at least as relevant and give other “main”-categories, for instance: in addition to the open vs closed systems, there are monoculture vs IMTA, fish vs shellfish, open sea vs coastal and freshwater vs seawater, to mention some. Given that fish farming systems is not the object of the review and that most, if not all, fish farming settings may benefit from the application of non-linear measurements of fish (and probably also of crustaceans and mollusks) behaviors, we think that expanding this subject in this section will take the focus out of the aim of the review. We hope that the addition of Figure 1, suggested by reviewer 1, can help to center the focus of the review and satisfy both reviewers.

Q5: Furthermore, I think it is necessary to specify this since the acquisition of good images takes place easily in the secondary type of structure compared to those at sea, where it is not yet possible or with many difficulties to be able to do so (i.e., sunlight, electronic issue, turbidity of water).

A5: We fully agree with the referee that the fish farming system used will greatly influence the selection of sensors for optimal signal acquisition. However, he/she must take into account that images constitute only one type of signal, and even though currently it is the most common one, that may not be the case in the near future as indicated in lines 55-71 and references 6 to 14 of the revised manuscript. Considering only images however, it must be borne in mind that it is possible to obtain images of excellent quality in sea cages in fjords for example (experience of the authors). In addition, given that fish move in 3D space, the signal should optimally provide 3D information of its behavior. In our opinion, sensors will be optimized and deployed that can provide 3D information on the fish behavior regardless of the quality of water in the near future. Indeed, use of images providing only 2D information have been shown to lead to false positive and false negative findings (see reference 106: Macrì, S.; Clément, R.J.G.; Spinello, C.; Porfiri, M. Comparison between Two- and Three-Dimensional Scoring of Zebrafish Response to Psychoactive Drugs: Identifying When Three-Dimensional Analysis Is Needed. PeerJ 2019, 2019, 1–25, doi:10.7717/peerj.7893). This subject in mentioned in the revised version of Future prospects and research needs, to which the text: “In our opinion, sensors will be optimized and deployed in the near future to provide 3D information on the fish behavior regardless of the quality of water” has been added in lines 854-856.

Q6: Line 77-78: It would be appropriate to quote other reviews and expanded this topic.

A6: While there are several and very good reviews using fish behavior as BEWS, we have not been able to find any review using entropy and fractal dimension (FD) analyses of fish individual and collective behaviors to set up Biological Early Warning Systems, except for the one we have recently submitted for publication. It would therefore not be appropriate, or ethical, to repeat the information contained in that review in the present work, that is the reason why we have clearly stated that the subject will not be dealt with here. However, to satisfy the referee’s comment, we have modified the text to include some key works on BEWS based on fish behaviors. Accordingly, the following text has been added (lines 80-83) in the revised version: . Changes in aquatic organisms’ behaviour have been proposed to serve as Biological Early Warning Systems (BEWS), to monitor for the presence of environmental contaminants in water resources [33–35] and for fish production in aquaculture [36–39].”

Q7: Lines 82-83: There are works where using CNN that can categorize behavioural patterns, such as foraging or aggressive interaction, and therefore already applicable or in implementation perspective to evaluate these behaviours (e.g., Yang, X., Zhang, S., Liu, J., Gao, Q., Dong, S., & Zhou, C. (2021). Deep learning for smart fish farming: applications, opportunities and challenges. Reviews in Aquaculture, 13(1), 66-90.). Therefore, authors must would discuss the advantages and disadvantages of such techniques…

A7: Even though, as already mentioned in the text and now clearly illustrated in Figure 1, the present review does not deal with the properties, pros- and cons-, of the different classifications methods , the original manuscript, as well as this revised version, already included 6 lines and 6 review references numbers 7 and 15 to 19)addressing this particular subject (lines 63-68). We are aware of works using different types of techniques and approaches to analyze behavioral patterns and that CNN is one of them. These techniques, however, contrary to the reviewer’s sentence (there are techniques…. already applicable or in implementation…) are not yet suitable to be commercially implemented, as mentioned in the abstract of the review the referee mentions: “…challenges still exist; DL is still in a weak artificial intelligence stage and requires large amounts of labelled data for training, which has become a bottleneck that restricts further DL applications in aquaculture.” Indeed, the need of reliable and comprehensive databases (already mentioned in our original version) is a major issue to successfully implement classification and IA techniques. Accordingly, we do not consider appropriate to include a discussion on techniques on classification, deep learning and AI techniques since that subject is not dealt with in our work and has already been amply covered by the recent and very good reviews already cited in the original version.

Q9: Therefore, authors must would ……. motivate how a non-linear dynamic approach could overcome this limitation

 A9: The referee does not say to which particular limitation needs to be overcome. Therefore, we cannot answer this question.

Lines 95-97 and paragraph 1.1:

I have two major points that authors should be considered.

Q10: For what my concern, it is missing an explanation of the reason why social hierarchy negatively influence individuals. In natural population, social structures are present and widely share among species because few individuals control resources. However, these structures are dynamic and change over time within and between generations, consequently populations have evolved and adapted to such conditions. For example, bigger individual control food resources but allowed smaller individual to exploit them. Thus, social structures per se do not cause any form of pain, at least when external condition are dynamically changing. One of the major limitation in aquaculture facilities is the predictability of resources that accentuate such aggressive behaviours and maintain them over time, thus few fish access to the resource and can grow while many others do not. By intervening on this aspect, farming animals’ conditions have seen an improvement on their survival rate and, in general, welfare. Studying the hierarchies within the groups in farms can help to understand the conditions of the group (see previous comment), but there are other aspects to consider that require relatively little effort.

A10: We cannot satisfy the need to explain the reason why social hierarchy negatively influence individuals because nowhere in the text is it indicated that social hierarchies negatively influence individuals or that social structures cause any form of pain. All the studies mentioned in the review are useful to point out to interrelationships between fishes and elucidate hierarchies which, as the referee and some experimental works indicate may change with time. The referee does not explain either what are: “other aspects to consider that require relatively little effort”; therefore, we cannot address that point either. In any case, the monitoring of hierarchies under farming settings does not exclude the use of any other aspect, particularly if they require little effort and provide valuable information.

Regarding the commentOne of the major limitation in aquaculture facilities is the predictability of resources that accentuate such aggressive behaviours and maintain them over time, thus few fish access to the resource and can grow while many others do not. By intervening on this aspect, farming animals’ conditions have seen an improvement on their survival rate and, in general, welfare” the reviewer is correct, and indeed modifications in the feeding schedule of farmed specimens (some studies are decades old) have shown an improvement in general welfare and fish growth, but, again, that subject is not the target of our work.

Q11: There is no ecological motivation on the importance of studying these dynamics and how they can lead to improvements, from a morphological, behavioural and cognitive point of view, of farmed fish (this is described briefly in the scope, but I would expand the discussion here). Authors should motivate this aspect.

A11: There are in fact no data indicating that “there is no … importance of studying these dynamics and how they can lead to improvements, from a morphological, behavioural and cognitive point of view, of farmed fish”. However, the are data indicating that social structures do exists and that there is a large lack of information on how the structures function under farming settings. Those studies are necessary to assess their potential impact on welfare for example, one way or the other: for instance, fish may feel better and more protected (higher welfare) in a safe, predictable shoal in which they know their position in the individual and collective interactions but they may also feel “worse” (lower welfare) if they have to compete for food with more dominant conspecifics in the shoal. To clarify this issue, we have added the following text in lines 93-96: “The fact that social structures and hierarchies under real farming conditions are largely unknown due to the difficulty to document them, does not mean that they are irrelevant or that they do not influence the health, welfare and other phenotypical/quality aspects of the production.”

Q12: The second point concerns the algorithm. Some software acquire information on two spatial dimensions, from which different types of behaviour can be extracted, including social behaviour, validated in numerous studies (e.g., Ethovision).

A12: Indeed, there are several programs used to obtain 2D information, of which Ethovision is one of them, idTracker and Kinovea are two other ones, and some authors have their own inhouse made software. We would like to stress again (see answer A5 and the text of the manuscript particularly reference 106 and lines 795-814) that 2D information is not optimal, since they may occasionally provide false positive and false negative findings.

Q13: In addition, supplementary analysis as factor analysis (PCA) of individually collected behaviours can provide an overview of individual and social behaviour. I would suggest adding a discussion on this point and what advantages it brings to use non-linear dynamics.

A13: A discussion on the advantages of the application of non-linear behaviors is already given in the original manuscript section 1.1, lines 143-155, and in the works included in the review, namely that non-linear analyses of behaviors have provided relevant information that classical analyses (regardless of whether the data are PCA treated) were not able to expose: i.e. they provide additional and relevant information.That is per se is a clear advantage for the farmer and for the scientist studying animal behavior. However, neither those authors nor us suggest that non-linear analyses are a substitute to classical analyses (nor the other way around either), as the referee seems to imply. To avoid this misunderstanding we have included a last sentence (lines 170-173) addressing this issue: “The reader must however understand that non-linear analyses of fish behaviors are not substitutes of classical methods. Rather, they must be considered providers of valuable additional information that, as mentioned above, classical analyses do not always readily expose”.

In addition, PCA analysis is one of several available tools that can clarify information provided by of both classical and non-linear analyses (i.e. it is fully possible to apply it to the results of non-linear analyses as well). It is not a surrogate to provide novel information when it is not contained in the original raw data submitted to the PCA-analysis.

Q14: Table 1: “school behaviour” has been widely used in respect to collective behaviour.

A14: Schooling behavior is one specific type of collective behavior that is different from shoaling behavior which is also a type of collective behavior. The term “schooling behavior” has been used correctly in the manuscript.

Q15: In addition, several works presented concern studies on single individuals (see “pain” section). Therefore, the review did not focus solely and exclusively on collective behaviour, and this aspect should be clarified.

A15: The referee is correct, the review is not focused solely and exclusively in collective behavior, and it is nowhere indicated that it is so. That is why we have included quite a few works deal with studies on single individuals and others deal with the behaviors of single individuals in groups in addition to the collective behaviors of groups (see also Table 1). To make this aspect even more clear from the beggining, we have changed the abstract of the revised version from the original “selected fish behaviors…” to “selected fish individual and collective behaviors…” (line 24).

Q16: Row 321: 3.5 Effect of the number of fish

A16: Apologies for this mistake, that has been corrected in the revised version.

Q17: Future perspectives and research needs: I would add a final paragraph on how these methodologies can be useful to producers, how they can learn and learn them, or what systems can be implemented for them to use (not all are experts and work with computations).

A17: The original manuscript already has a paragraph, as suggested by the reviewer, indicating which type of information the application of non-linear analyses of fish behaviors many have for the fish farmer (lines 805-815). As the referee is aware of, these methodologies are still too immature to be satisfactorily implemented under real production settings, particularly, but not only, due to the lack of suitable sensors and databases necessary to classify the behaviors. Therefore, it is premature to suggest what techniques/equipment the farmer “must learn and learn them, or what systems can be implemented for them to use”. Although we also would expect farmers not to be experts with computation, the developers of the solutions must be, and the tools they provide (both sensors and software for data acquisition, interpretation and classification) should be robust and easy to use. As said, that area is still under development. To comply with this request, we have added the following text: (lines 892-895): “However, and as already mentioned above, practical application and successful implementation of these techniques still requires a wealth of additional experimental and practical data obtained under real farming settings, under different conditions and for different species that allow the development or reliable ready-to-use industrial equipment.”

Round 2

Reviewer 1 Report

This is the second round of review, and I am pleased to see that the writing has improved significantly. The paper is of high quality and contributes to the scientific research. I suggest that the authors bring the paper up to publication quality. Here are my suggestions for the authors' final edition:

(1) Figure 1 (line 189): Consider increasing the font size in the Figure 1 to improve readability for readers.

(2) Appendix A (line 826): On page 5 of 34, please check the table border (the top line) for accuracy.

(3) Reference 1 (line 834): Ensure that the format of this reference is correct, and that readers can locate it easily on the internet.

(4) Reference 66 (line 1001): When referencing non-English sources, it is essential to work closely with the assistant editor to ensure that the paper can be found online.

Reviewer 2 Report

The revised version of the paper has greatly addressed my previous concerns. This is a nice and robust paper with an important perspective for future research.  I especially like the addition of Figur 1, which easily clarifies the aim of this study and possible application of the non-linear methodology for understanding animal behavior and, then, their state of welfare. I have one minor comment for figure 1 regarding the size and color of title pannel "target of this review" that the authors might want to reduce.  Overall, this is a strong and interesting paper.